# Similarity Metrics Enforcement in Seasonal Agriculture Areas Classification

**Marcio A. S. Santos [1],\***, **Eduardo D. Assad [2]**, **Angelo C. Gurgel [3]** and **Nizam Omar [1]**

1 Computing and Informatics School, Mackenzie Presbyterian University—Rua da Conslação, 930 Consolação, São Paulo SP 01302-90, Brazil; omar@makenzie.br
2 Embrapa Informatica Agropecuária, Environmental Modeling Lab—Av. Andre Tosello, 209 Unicamp Campus, Campinas SP 13083-886, Brazil; eduardo.assad@embrapa.br
3 Getúlio Vargas Foundation/São Paulo School of Economics—Rua Itapeva, 474 Bela Vista, São Paulo SP 01332-000, Brazil; angelo.gurgel@fgv.br
* Correspondence: marcioassantos@live.com or 71606815@mackenzista.com.br; Tel.: +55-(19)-99113-6236

**Abstract:** Accurate identification of agriculture areas is a key piece in the building blocks strategy of environment and economics resources management. The challenge requires one to deal with landscape complexity, sensors and data acquisition limitations through a proper computational approach to timely deliver accurate information. In this paper, a Machine Learning (ML) based method to enhance the classification process of areas dedicated to seasonal crops (row crops) is proposed. To this objective, a broad exploration of data from Moderate Resolution Imaging Spectro-radiometer sensors (MODIS) was made using pixel time-series combined with time-series similarity metrics. The experiment was performed in Brazil, covered 61% of the total agriculture areas, five different states specifically selected to demonstrate biome differences and the country's diversity. The validation was made against independent data from EMBRAPA (Brazilian Agriculture Research Corporation), RapidEye Sensor Scene Maps. For the eight tested algorithms, the results were enhanced and demonstrate that the method can rate the classification accuracy up to 98.5%, average value for the tested algorithms. The process can be used to timely monitor large areas dedicated to row crops and enables the application of state of art classification techniques, two levels classification process, to identify crops according to each specific need within the areas.

**Keywords:** remote sensing; agriculture; time series similarity metrics; machine learning; land use dynamics

---

## 1. Introduction

The identification of agriculture areas is valuable information for the scientific community, government agencies, farmers, and other members of the society. Agriculture main commodity areas are monitored on a global scale to predict production, yield, demand, prices, climate risks, and so forth [1–5]. Related statistics are produced and published to reduce the economic externalities impact and to balance market information asymmetry [6,7]. Ongoing, accurate, and timely crop information remains a challenge, as data becomes available long after the harvesting time, and the machine learning classification is dependent on reliable data sources and processing needs enhancements, despite the available capacity [8–10]. The accuracy of the latest results in distiguishing crops of agriculture areas range from 84% to 95% [11–14]. The state of the art publications in agriculture remote sensing explore the importance of machine learning applied in combined data sources and types in order to enhance the process and accuracy for multiple crops as classes.

When it comes to process improvements and machine learning, Convolutional Neural Networks (CNN) was applied to achieved significant results in image recognition tasks by automatically

learning a hierarchical feature representation from raw data, and through combinations of time series (temporal dimension) with 2D texture images (spatial dimension) to enhance features that could not be gained in a single dimension. The unified frame work, times series, and CNN, demonstrated competitive accuracy when compared with the existing deep architectures and the state-of-the art time series algorithms [15]. The exploration of synergies between different sources of data to improve classification of high-level spatial features produced by hierarchical learning (i.e., scene labeling) to contrast with low-level features such as spectral information (morphological properties) has also been made. Temporal and angular features played more important roles in classification performance, especially abundant vegetation growth information. Multispectral and hyper-spectral fusion successfully discriminated natural vegetation types diversity [16]. Also using fusion methods techniques, in this case to mitigate spatiotemporal limitations of multi and hyper-spectral data from multiple sensors, a compatibility between the hyper-spectral data and Sentinel-2 (multi spectral) data has been validated. The method opened new possibilities for classifying complex and heterogeneous land covers in multiple environments with the combination multiple data sources [17]. On the other side, when accuracy enhancement is the major focus, an automated mapping process for soybean and corn using crop phenology characteristics with time series and topographic features from multiple sources combined was proposed by Reference [18]. The classification achievements range from 87% to 95%, which is an increase of 2.86% on previous published works [19]. For the case of row-crop areas in Brazil, as is the case of this work, international publications that addresses this need were observed. A experiment that covered 3 crops (corn, soybeans and cotton) in Mato Grosso, used time series correlation coefficient and successive classifications has been performed to detect agricultural areas. The approach was capable to achieve 95% accuracy and kappa index of 0.98 [14]. In another study, a rigorous multiyear evaluation of the applicability of time-series for crop classification in Mato Grosso was performed. The conclusion showed progress in refined crop-specific classification and appointed the need for grouping of crops as classes. The results were consistently near or above 80% accuracy and Kappa values were above 0.60. The authors also highlight the need for additional research to evaluate agricultural intensification and extensification in this region of the world [20]. The combination of different channels (red and near-infrared) were explored with five algorithms (Maximum Likelihood, Support Vector Machines, Random Forest, Decision Tree, and Neural Networks). The methods accuracy ranged from 85% to 95%, and demonstrated that 250 m imagery is efficient to map fields down to 20 ha. Results also suggested that cropland diversity could be addressed using regional and specific landscapes training sets [13]. As a reference, the accuracy assessment of a supervised classification on Landsat 8 satellite images was performed. The results indicated that the object classification was better than the classifications by pixel and the best thematic map was generated by the SEGCLASS classifier. The accuracy achieved was 74% and kappa index 0.57 [21].

In common is that References [11–13,18,21–24] share the best results in the crop classification with their methodologies within the studied area, and present the challenges related to the limited amount of training data to scale up the process. As each of the studies was performed within unique field information and conditions, including different crop varieties contrasted, the results cannot be compared properly.

In this research the aim is to propose a process to enhance classification accuracy in agriculture areas dedicated to row crops and put them in evidence to support the land monitoring process and generate a first knowledge layer to support specifics needs and fine-tuning classification. Although time series has been extensively explored to classify agriculture and it is a well-stablished process [25,26], the combination of time series and similarity metrics to explore the classification as proposed is a novel, explore the crop growing season cycle of a variety of temporary crops with multiple algorithms.

### 1.1. Remote Sensing

The use of the pixel as a sensor in remote sensing to monitor agriculture cycles requires quality images and fine specs to be granularly classified accordingly to each unique purpose, so unexpected

dynamics can be detected and properly managed [8,9], (e.g., crop phenology stage, chemicals administration, mechanization, and irrigation management, among others). The critical components for remote sensing are: accurate and current information for training the classifiers; an affordable source of data that qualifies for the specific objective; a storage and processing capacity [10,27,28]. By using MODerate-resolution Imaging Spectrometer (MODIS) products that include the Enhanced Vegetation Index (EVI) [29], among others, we have access to a complete record of data from each of the Terra and Aqua MODIS sensors, at varying spatial (250 m, 0.05 degree) and temporal (8-day) resolutions validated with accuracies depicted by a pixel reliability flag and with globally averaged uncertainties of 0.015 units. Further, the MODIS/EVI combination is a robust set for exploring seasonal crops (soybeans) [22]. The spatial resolution adopted are consistent in expressing accurate cropland information in fields that are larger than 20 ha [13], and appropriate to this task [30]. This experiment relies on the achievements and specifications above to explore the spectral dynamics of extensive areas, in accordance with the specifics purpose stablished for this research.

### 1.2. Brazilian Agriculture in Numbers

In Brazil, agriculture represents almost a quarter of the country GDP's, 24.1% in 2017 [31]. According to the Brazilian Institute of Geography and Statistics (IBGE) rural census [32], the harvested area during the 2016/2017 season represented 7.9% of the total area with 73,797,057 hectare (ha) dedicated for temporary and semi-temporary crops and another 5,184,813 ha for permanent crops, with the final results published in 2019. For the purpose of this study, detailed statistics of the studied areas are summarized on the Table 1. The column "Total" shows the total area (country/State). The column "Agriculture" presents the harvested area (country/State). It is important to highlight that, due to geographical and environmental conditions, which include climate and/or technologies (e.g., irrigation), it is common to have multiple crops per year in tropical areas. Therefore, the conclusion is that the harvests are larger than the total area dedicated to agriculture, a total of 67,547,537 ha according to EMBRAPA [33,34] and MAPA [35].

The main crops cultivated locally are presented below, in Table 2, and the data is organized as a percentage of total area occupied to provided a land use perspective view, and put in evidence of a larger area per crop cultivated.

**Table 1.** Studied areas (ha).

| Country/State | Total | Agriculture | (%) |
|---|---|---|---|
| Brazil (BR) | 851,605,394 | 67,547,537 | 7.9 |
| Goias (GO) | 34,011,178 | 6,106,279 | 17.9 |
| Minas Gerais (MG) | 58,652,212 | 4,814,438 | 8.2 |
| Mato grosso (MT) | 90,336,619 | 14,872,045 | 16.5 |
| Paraná (PR) | 19,930,792 | 8,776,871 | 44.0 |
| São Paulo (SP) | 24,822,362 | 6,835,741 | 27.5 |
| State Subtotal | 227,753,038 | 41,405,374 | 61.3 |

Source: https://censos.ibge.gov.br/agro/2017/.

**Table 2.** Temporary crops.

| Crop Type | Area (ha) | Fraction of Total (%) |
|---|---|---|
| Soybean | 30,622,460 | 45.3% |
| Corn | 17,985,764 | 26.6% |
| Sugar cane | 9,153,709 | 13.6% |
| Beans | 3,069,622 | 2.1% |
| Wheat | 1,796,065 | 2.6% |
| Rice | 1,778,190 | 2.6% |
| Manioc | 943,323 | 1.4% |
| Cotton | 910,057 | 1.3% |
| Total area | 64,660,183 | 93.7% |

Source: https://censos.ibge.gov.br/agro/2017/.

## 2. Methods

### 2.1. Studied Area

The study was performed over Brazil at the state level, see Figure 1. The locations were selected to bring the diversity (biome, crop, climate, etc.) and its complexity into the analysis. The states studied were: Goiás (GO), Minas Gerais (MG), Mato Grosso (MT), Parana (PR) and São Paulo (SP), Table 1. In this study the growing season (phonological) cycle is the key aspect taken in consideration, crop recognition requires specifics and may be performed in accordance to them, combining scenes and the growing season maps.

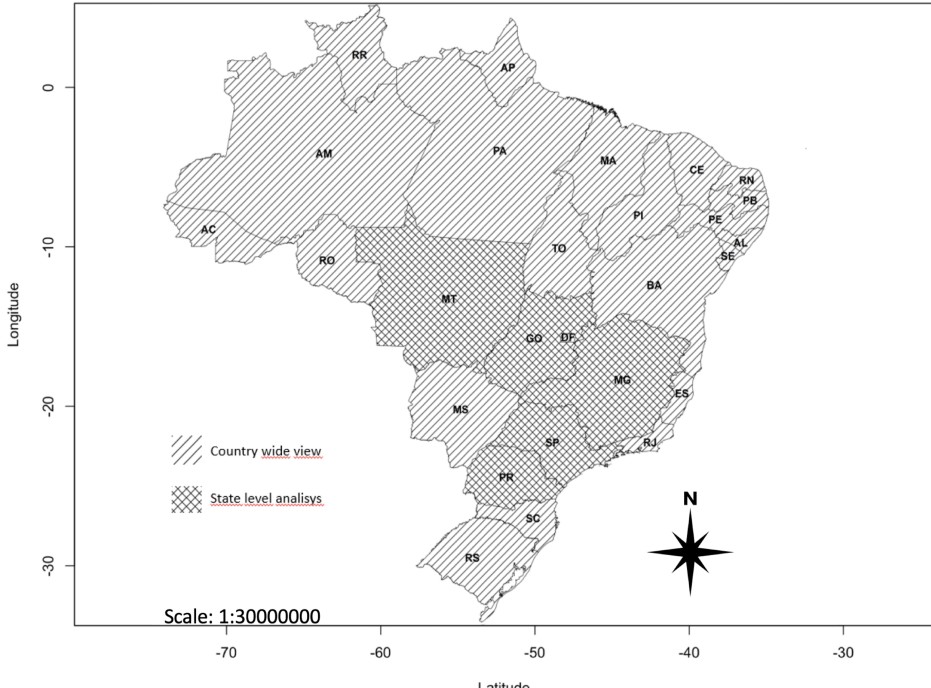

**Figure 1.** The experiment was performed over GO, MG, MT, PR, and SP, and a country-wide view.

### 2.2. IBGE Census

As a reference in this study, IBGE Agriculture census data was used to contrast results, that is, quantify the total area and row crop areas for each studied state. The original numbers were consolidated and published by municipalities and by state, consequently. For this research objective the productive clusters are represented in maps and at the state level, without municipal geopolitical boundaries. The preliminary census results were published in 2018 and final results late in 2019.

### 2.3. Ontology

The use of expert knowledge management based on ontology process to support the collective understanding of a single event (*Data driven approach*) was applied to proper address crop growing season specifics to expand the comprehension possibilities in a multidisciplinary context [36]. The same author defined ontology scope as being the "specialist of specifics", meaning the understanding of a domain group. In this environment the essential is the potential for information and knowledge sharing [37]. The proposed Ontology is presented in the concept map below, Figure 2, where the target class are highligted and the profiles are demonstrated. The data was segmented in accordance to the Global Food Security (GFS) [38] and MODIS ontology into two large classes, Agriculture and Other. Agriculture, as the target class, included: Irrigated and dry seasonal crops areas potentially featuring corn, soybean, cotton, barley, potato, alfalfa, sorghum, rye, canola, peanut, manioc, and beet.

The Others class included: perennial and semi-perennial/semi-temporary crops, natural vegetation (forest, cerrado, amazon forest, tropical forest), pasture, urban areas, and water surface, among others. The agriculture class areas are those where we expected to find a high level of EVI values dynamics, which is the purpose of this study, and is also compatible with the objective of the sensor selected. As a reference, the profile images that are representing the target group are from SatVeg [39]. Only the crops that can be found within the region and are described as part of sensor ontology are here in evidence.

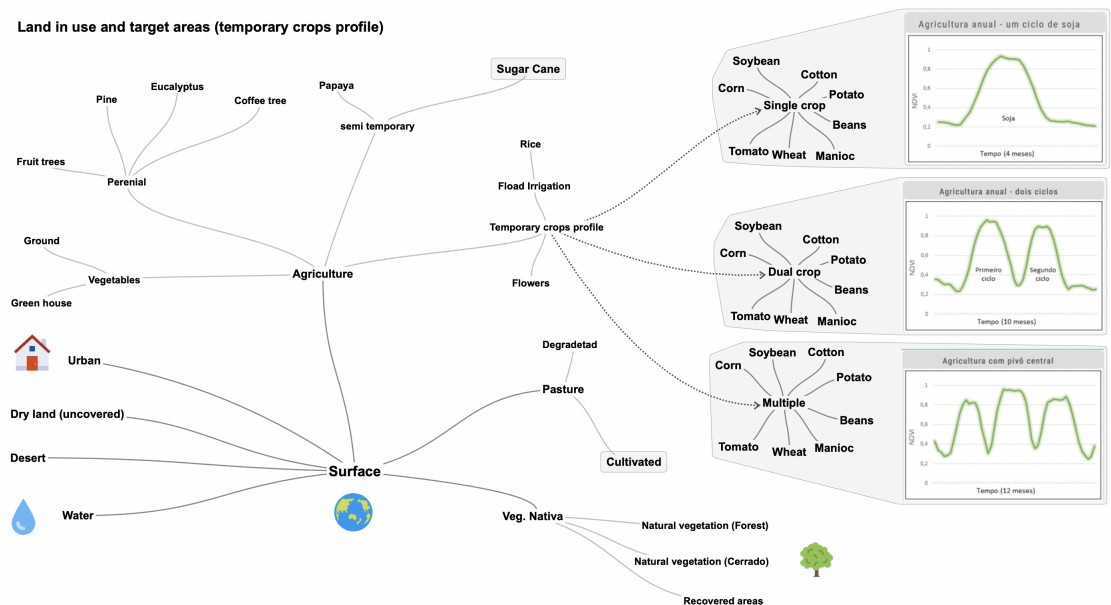

**Figure 2.** A concept map of the ontology used—Target groups highlighted above share seasonal characteristics with each other and within the time frame selected from EVI collected maps. Sources: EMBRAPA, MODIS, IBGE.

### 2.4. Datasets Processing

The criteria for the selection of the sensor was based on the most recent publications and goal, as cited above. The .tiff (Tagged Image Format) dataset files were downloaded from the MODIS repository as .hdf (Hierarchical Data Format) files according to provided guidelines, and then framed in accordance to each studied state and a country view. The experiment was performed using EVI from MOD13Q1 with 250 m of spatial resolution.

Accord to the methodology outlined as a flowchart in Figure 3, a large amount of random Pixels Time Series (PTS) were extracted from each studied area to achieve $99\%^{+1}_{-1}$ of confidence level, with 17,000 points for each dataset, and 5 datasets per state. PTS values were extracted as vectors and stored as .csv (Comma Separated Values) files using a expert system created for this purpose. Each vector contained the pixel position, EVI raw data as time series values, and calculated computational distances, with similarity metrics proposed. The data were then pre-classified into two large groups, Agriculture and Others. Agriculture was the target group, which expected areas with high spectral dynamics level, and the Others group contained lower dynamics level areas. Each experiment used 20% of data set for training.

Given the inherent complexity of vegetation and environments which are all reflected within the region selected, for each state, a group of similarity metrics were selected and tested as a way to put in evidence the expected characteristics and work as a hybrid index in the classification process, as in Reference [40]. The computational distances used in this experiment are: Manhattan distance, Minkowski distance, Sum, Mean, Median, Standard deviation, Coefficient of Variation, Variance, and Difference [41,42]. All the similarity metrics that used the distances were calculated for each pixel time-series. In particular, the Minkowski distance used has been explored by References [43,44] to

cluster time series that have different temporal resolutions, as the case for the Dynamic Time Warp (DTW), and in this case it was used to measure the length of the time-series. The possible values for the *c* are useful to accommodate the time difference into the time series. The equation is demonstrated below, Equation (1):

$$D(p,q) = \left(\sum_{i=1}^{n} |p_i - q_i|^c\right)^{1/c}. \tag{1}$$

where: the distance $D$, $p$ and $q$ are the data points ($x_{i+1}$ and $x_i$, respectively) and used $c = 2$.

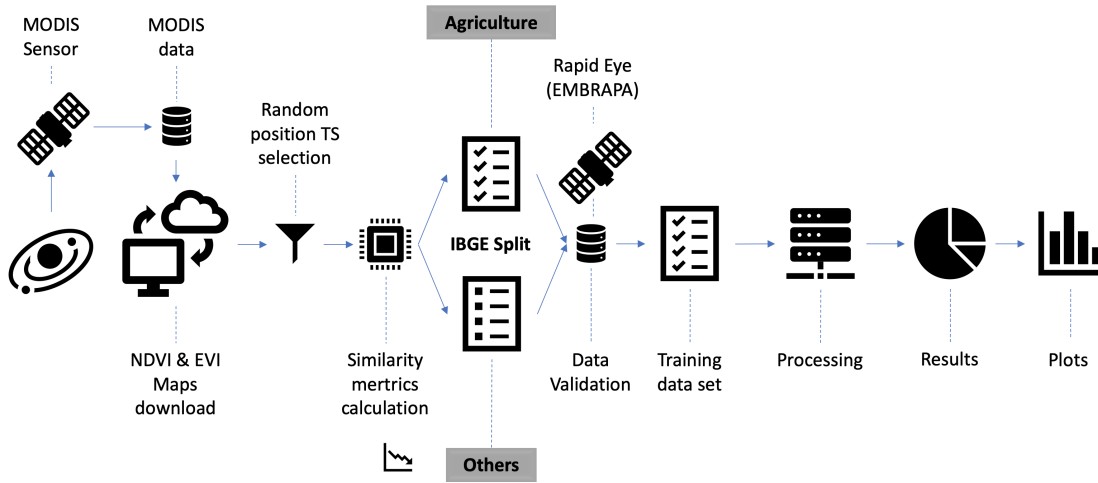

**Figure 3.** Process Flow—Target groups selected—Agriculture & Other. Elaborated by the authors.

### 2.5. Time-Frame

For this research, 36 raster layers from the MODIS sensor were used, with an 18 month time frame coverage, from July 2016 to December of 2017 to match IBGE census data also used as a reference in this study. The time frame selected meets the objective of measuring annual dynamics for temporary crops, thus excluding semi-temporary crops (e.g., sugarcane) from the analysis.

### 2.6. Validation

Tree levels of validation were considered—the classification accuracy, the concordance between algorithms, and the computational metrics relevance. EMBRAPA scene maps were used as an external reference data source to contrast and validate accuracy. The spatial resolution of the the validation maps was 5 m to provide quality data for the training set used, to preserve the characteristics that are relevant to the experiment [8], and to match the field size.

The classification accuracy validation was made according to the proposed ontology, a confusion matrix used to contrast results as True-Positive (TP), False-Negative (FN), False-Positive (FP) and True-Negative (TN). Accuracy stands for all that were correctly classified, as in the Equation (2), and recall stands for positives that were correctly classified, as in the Equation (3), below:

$$accuracy = \frac{TP + TN}{TP + FP + TN + FN}, \tag{2}$$

$$recall = \frac{TP}{TP + FN}. \tag{3}$$

The concordance between algorithms assessment were made using the Cohen's Kappa coefficient or simply (*k*), which has been largely used to evaluated classification algorithms performance as described in Reference [45]. Kappa values ranges from −1 to 1, where −1 represents "complete disagreement", 0 is a "random classification", and 1 is a "perfect agreement".

The computational metrics used are the sensitivity and specificity, Equations (4) and (5) below, where we expect to identify the impact of the selected similarity metrics composition on the classification process of the target group.

$$sensitivity = \frac{true\,positive}{positive} \tag{4}$$

$$specificity = \frac{true\,negative}{negative}, \tag{5}$$

where "true positive" is the number of correctly predicted areas, "positive" is the number of agriculture, "true negative" is the number of correctly predicted as others, and "negative" is the number of others shown in the classification.

The impact of each similarity metric as an attribute over the target class was evaluated using Shannon entropy index to reveal in a [0:1] scale the cluster behavior [46,47]. The calculation of the distances was made according to the Equation below, Equation (6):

$$E_i = -\sum_{i=1}^{n} p_i \log(p_i), \tag{6}$$

where $E$ is the information strengh of the attibute $n$ over the class $i$, and $p$ is the probability of the class in $n$.

### 2.7. Computation Tolls

All data manipulation and tests were performed using $R^{\circledR}$ language, *Matlab$^{\circledR}$*, *RapidMiner$^{\circledR}$*, and *QGIS$^{\circledR}$*. Using these tools to evaluate the effectiveness of the proposal, we explored the data with eight Machine Learning algorithms: Naive Bayes, Logistic Regression, Decision Tree, Gradient Boosted Tree, Generalized Linear Model, Deep Learning, Random Forest. This collection represents most of the algorithms used in the cited publications.

The data used were min-max normalized to comply with the classification tolls specs. The values were linearly reduced to a scale between [0:1], where 0 and 1 are the minimum and maximum values, respectively. The $z$ is the normalized value accord to Equation (7) below:

$$z_i = \frac{x_i - min(x)}{max(x) - min(x)}. \tag{7}$$

### 3. Results

By performing the study as proposed, we observed that the addition improved the specificity and sensitivity of the algorithms. The sensitivity of the algorithms can be verified by contrasting ROC comparison curve (detection probability in machine learning). The Figure 4 shows the classification process for data without similarity metrics and Figure 5 shows the sme process for data with similarity metrics. Specificity raised from 98.6% to 99.2% with a direct effect on accuracy, sensitivity enhancement from 94.62% to 97.4% which improved processing performance despite the data dimensional increase. As we can verify, data with similarity metrics increased sensitivity for all tested algorithms.

The enhancement for each method is demonstrated below, Table 3, where the results for each state are presented. The caption MT, GO, MG, SP and PR are the abbreviations of the states mentioned on Section 2.1. The columns RD and DSM represent the accuracy with the different type od data, Raw Data (RD) and Data with Similarity Metrics (DSM). It is important to point the difference in results between the algorithms, these differences reflect the way each of them internally organizes the data toward the best solution.

The concordance of the algorithms index (Kappa) and the enhancement range achieved for each method with Raw Data and Data with Similarity Metrics (RD-DSM) is demonstrated in the Table 4.

The highest impact was achieved with Deep Learning, increased the accuracy from 83.9% to 98.6% with Kappa index of 0.95. However, the highest accuracy was achieved with Gradient Boosted Tree, results were enhanced from 95.2% to 99.6% and with Kappa index of 0.96.

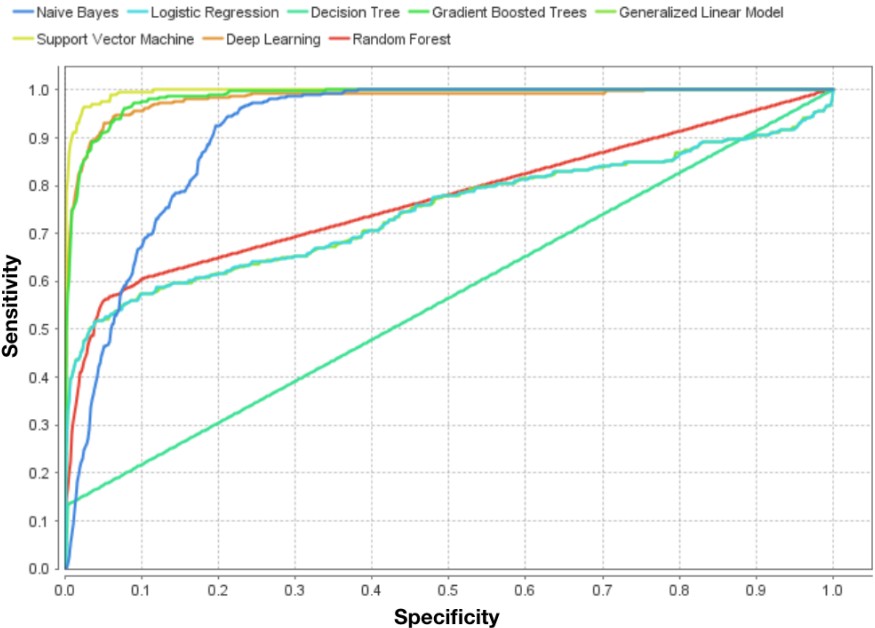

**Figure 4.** ROC—Classification process effectiveness—Data witout similarity metrics on the data set. Comparison—Naive Bayes, Logistic Regression, Decision Tree, Gradient Boosted Tree, Generalized Linear Model, Deep Learning, Random Forest.

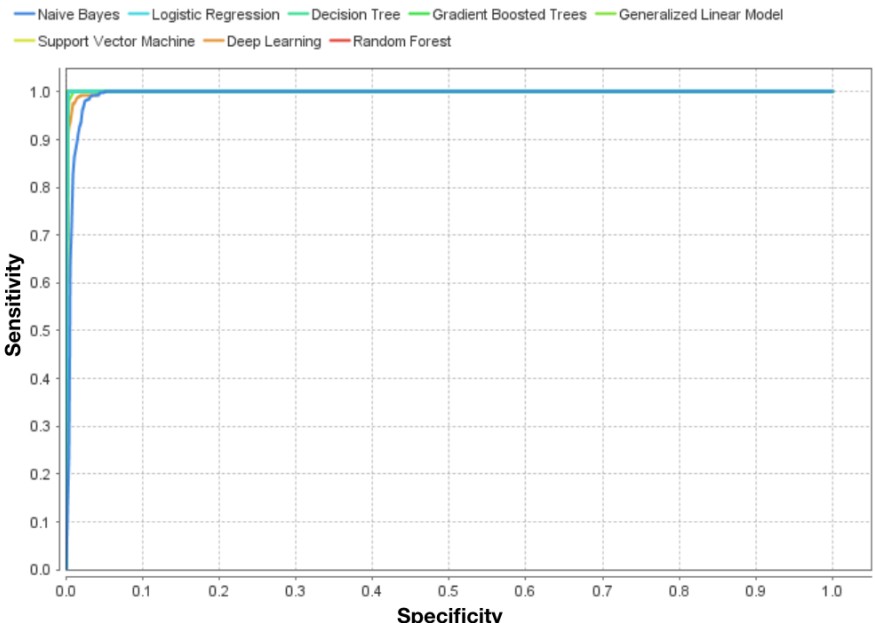

**Figure 5.** ROC—Classification process effectiveness—Data with similarity metrics as part of the data set. Comparison—Naive Bayes, Logistic Regression, Decision Tree, Gradient Boosted Tree, Generalized Linear Model, Deep Learning, Random Forest.

The hierarchal clustering analysis below, Figure 6, put in evidence the hierarchal importance of each metric for the process. The relevance of each attribute is calculated accord to Shannon information and the normalized score is represented hierarchically. The Minkowski distance played a key part in

the classification process and demonstrated to be very representative of the target class, highest score. Canberra distance, Standard deviation, Mean, Coefficient of Variation, and Variance were also very relevant in the high score process, but not as decisive. The removal of the Manhattan distance, Sum, Median, and Difference had no impact on the process accuracy.

**Table 3.** Accuracy classification results.

| Algorithms | GO | | MG | | MT | | PR | | SP | |
|---|---|---|---|---|---|---|---|---|---|---|
| | RD | DSM | RD | DSM | RD | DSM | RD | DSM | RD | DSM |
| Naive Bayes | 88.4 | 94.5 | 84.2 | 93.8 | 85.8 | 93.1 | 89.2 | 94.1 | 80.0 | 90.5 |
| Generalized Linear Model | 93.4 | 98.4 | 96.9 | 99.3 | 94.9 | 99.3 | 91.1 | 98.4 | 89.7 | 97.8 |
| Logistic Regression | 97.4 | 97.4 | 97.7 | 98.6 | 97.4 | 98.9 | 91.0 | 98.7 | 96.3 | 97.8 |
| Deep Learning | 83.9 | 97.9 | 86.9 | 99.2 | 88.9 | 99.2 | 85.1 | 98.2 | 83.9 | 97.6 |
| Decision Tree | 85.2 | 98.6 | 92.3 | 99.4 | 92.3 | 99.1 | 85.1 | 98.7 | 86.5 | 97.5 |
| Randon Forest | 85.8 | 98.2 | 92.3 | 99.3 | 95.8 | 99.6 | 86.9 | 98.9 | 76.2 | 97.6 |
| Gradient Boosted Trees | 96.0 | 98.4 | 97.1 | 99.4 | 97.4 | 99.6 | 96.0 | 98.8 | 95.2 | 97.8 |
| Support Vector Machine | 97.9 | 98.3 | 98.1 | 99.5 | 97.8 | 99.1 | 97.4 | 98.9 | 97.0 | 92.2 |

**RD**: Raw data and **SM**: Data with Similarity metrics.

**Table 4.** Concordance.

| Tested Models | (RD-DSM) | Kappa |
|---|---|---|
| Naive Bayes | 80.0–93.8% | 0.89 |
| Generalized Linear Model | 89.7–99.3% | 0.97 |
| Logistic Regression | 91.0–99.3% | 0.96 |
| Deep Learning | 83.9–98.6% | 0.95 |
| Decision Tree | 86.2–99.4% | 0.94 |
| Randon Forest | 86.6–99.5% | 0.97 |
| Gradient Boosted tree | 95.2–99.6% | 0.96 |
| Support Vector Machine | 97.0–99.5% | 0.95 |
| **Average** | **88.7–98.6%** | **0.95** |

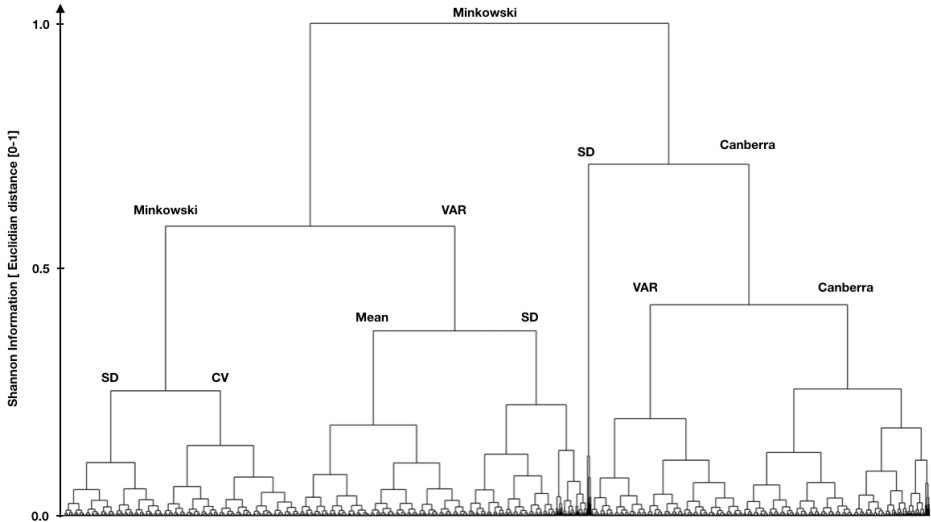

**Figure 6.** Shannon information with Euclidian distance for the key main decision levels. In this case: Minkowski Distance (Minkowski); Coefficient of Variation (CV); Standard Deviation (SD); Canberra distance (Canberra); Mean value (Mean).

The correlation between a group of indexes found in the bibliography were also evaluated as alternatives to the EVI. Results are presented in Table 5, where the correlations are contrasted. From this perspective, none of the indexes were found as strong enough to be used as alternative or to be replaced within the same method.

**Table 5.** Correlation.

| Correlação | EVI | NDVI | NIR | MIR |
|---|---|---|---|---|
| **EVI** | 1 | | | |
| **NDVI** | 0.1964 | 1 | | |
| **NIR** | 0.6510 | 0.1088 | 1 | |
| **MIR** | 0.3227 | 0.3283 | 0.2417 | 1 |

Finally, from the process it was possible to generate a scene map layer identifying row crops areas, as presented on Figure 7, below. The figure shows where the row crops clusters are located.

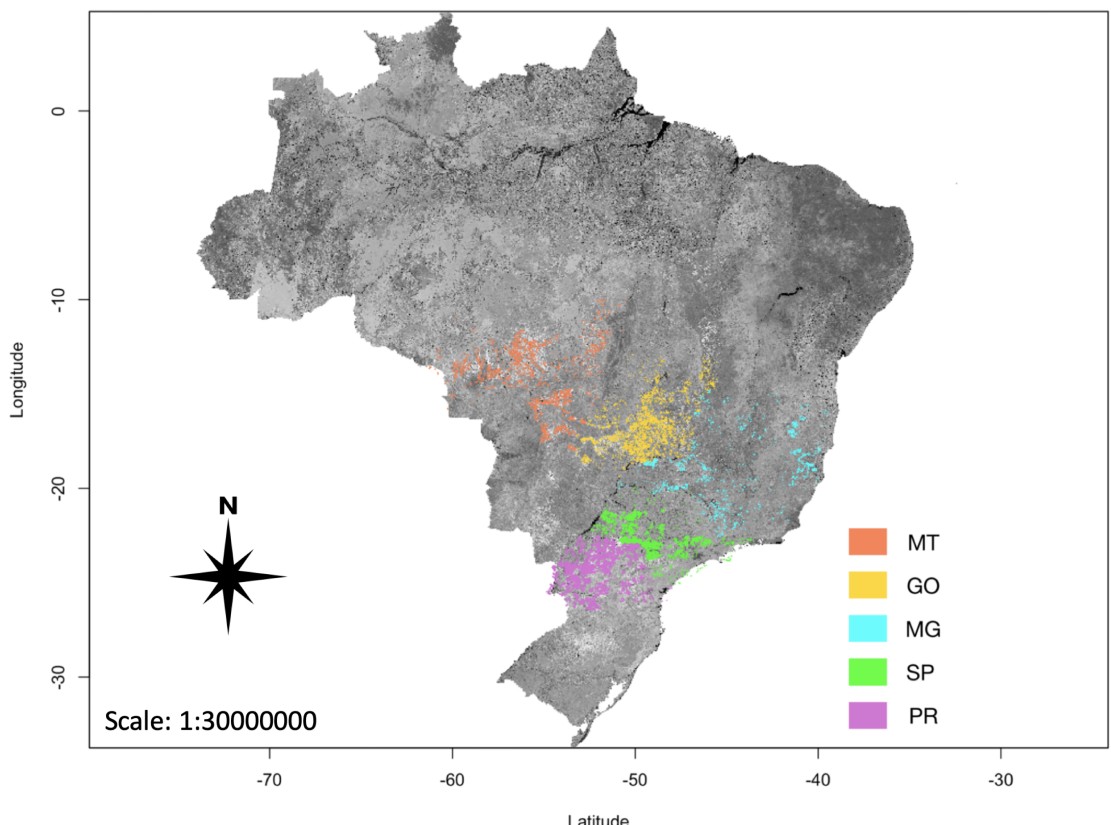

**Figure 7.** Country view without geopolitical boundaries.

## 4. Discussion

As presented in the Results section, the proposed approach enhanced the classification accuracy for all tested methods. The similarity metrics in combination with time series improved the average accuracy from 88.7% to 98.5%, as presented on Table 4. Therefore, the proper selection of similarity metrics showed potential to enhance the classification efficiency, increase classification accuracy without extra processing time.

The results achieved are higher than all previous work that addressed classification of agriculture row crop areas in Brazil and, in particular, the results achieved by References [13,14,20–22,24] as presented in Table 6. In regard to Reference [14], which achieved the highest result prior to this work,

despite the large area, they only study one state (Mato Grosso) and tree crops. The second highest result achieved [13], compared results for 6 algorithms and results ranged from 84% to 96%, covered 5 countries and noticed the lowest accuracy for Brazil (84%). The same author appointed as a key finding that "the site effect dominates the method effect".

**Table 6.** Results contrasted.

| Author | Area (ha) | Accuracy | Kappa | Comments |
|--------|-----------|----------|-------|----------|
| Publications that share at least one common aspect: Brazil as studied area | | | | |
| [14] | 5,617,250 | 95.0% | 0.98 | - 3 tree seasonal crops in Mato grosso/ |
| [13] | 507,728 | 84.0% | | - used the same algorithms, achieved 84% accuracy for Brazil/ |
| [22] | 724,293 | 84.0% | 0.56 | - differentiated Soybean and non-soybean/ |
| [24] | 1600 | 83.0% | 0.78 | - used high spatial resolution images/ |
| [21] | 9658 | 74.6% | 0.57 | - seasonal crops group as part of the results/ |
| International Publications that share at least one similarity | | | | |
| [48] | 258,500 | 97.0% | | - seasonal crops group as part of the results/ |
| [12] | 2,800,000 | 94.6% | | - seasonal crops classification/ |

It is important to highlight that in Brazil we have summer crop season which is in general rainy and cloudy, more than 90% of the production is rainfed. According to References [49,50], less than 40% of temporal images are useful for analysis. In this context, in this research, we bring the need for the combination of high temporal resolution with high spatial resolution for validation, MODIS and RapidEye, respectively.

According to Reference [48] the combination of temporal and spectral information can improve classification accuracy than only using spectral information, 10–15% higher. In this work, we demonstrated that the proper similarity metric can enhance time series classification accuracy, 3–14%, 10% on average for the tested algorithms. In fact, the combination created condition to increase specificity (accuracy) and sensibility (convergence). Both aspects are demonstrated on Figures 4 and 5 above. The sensibility, in this case the positive impact on the classification process, for each of the algorithms were different. Below we present the sensibility for each one of the algorithms:

- Low: Suport Vector Machine, Deep Learning, Gradient Boosted Tree
- Moderate: Naive Bayes
- High: Logistic Regression, Decision Tree, Gradient Boosted Tree, Random Forest

Two level classification strategy, as proposed by Reference [18], or multi-level as used by [14] are a possibility from the achieved results. The layer identifying temporary crop clusters, as presented on Figure 7, meets the demand presented by References [12,20,22,48], among others.

Another important aspect is the potential of the process to remotely identify agricultural areas as a dynamic census process. Timely spatial information about productive areas with a high confidence level ($99\%^{+1}_{-1}$) is relevant for policy makers and for the private sector. This information can support IBGE statics and contribute to their monitoring process. In regard to private sector, precision agriculture services can be provided in large scale when geospatial data organized and available.

The process also demonstrated some limits that are related to the objective of each search and complexity of the environment. An accurate time-frame collection of maps is required to enhance the uniqueness of the target. It is also important to remember that on top of spectral resolution, temporal and spatial resolutions are key to put in evidence specifics of cultivars. Therefore, sugarcane and cultivated pasture have specifics in growing season, and requires specifics in spectral and temporal resolution to enhance the classification process [51].

## 5. Conclusions

This study explored the classification of agriculture areas dedicated to temporary crops using well-known ML algorithms to examine time-series from a collection of vegetative indexes in combination with similarity metrics. As a conclusion we have that:

- As the primary objective, the results demonstrated that the approach enhanced the classification accuracy for all tested algorithms. This is the highest accuracy for the classification of agriculture areas in Brazil, 99.6% with kappa index of 0.96 using Gradient Boosted Tree algoriothm;
- The similarity metrics worked to increase the accuracy within the context proposed, EVI data reflecting the growing season dynamics of temporary crops. The similarity metric added 3–14% points to the accuracy;
- The process increased accuracy and without extra computational cost.
- The results are robust to support policy maker and precision farming, $99\%^{+1}_{-1}$ of confidence level.

The field knowledge (scene map generated) allows crop level classification improvements, that is, crop differentiation using the available techniques accordingly to each specific need. This information is useful for further research and also to support the private sector and public sectors on monitoring and spatial planning of annual crops in Brazil.

To enhance results and explore the approach, modifications in the Minkowski equation, *c* value, should be tested to address specifics in phenology time frame. The size of the data set for training should be quantified, it is expected that the a smaller data set size would maintain accuracy. Moreover, DTW can be used to explore the differences in growing season for the two levels crop classification process, among others. Specifics in computational costs and cloud computing benefits should also be explored and demonstrated.

**Author Contributions:** M.A.S.S.: Conceptualization of this study, Methodology, Software coding and manipulation, Validation Methodology, Writing—Original draft preparation; E.D.A.: Review of the methodology and validation process; A.C.G.: Writing review; N.O.: Supervisor. All authors have read and agreed to the published version of the manuscript.

**Funding:** This research received no external funding.

**Acknowledgments:** We thank Mackenzie Presbyterian Univesity, Embrapa Agroinformatica, Fundação Getúlio Vargas, Capes, for the support with this research. We also thank Vanessa Pugliero and Eduardo Pavão from Embrapa for the support with organization of the information used during the validation process.

**Conflicts of Interest:** The authors declare no conflict of interest.

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
