# Peer review of "Similarity Metrics Enforcement in Seasonal Agriculture Areas Classification"

_remotesensing, doi:10.3390/rs12111791_

Round 1
Reviewer 1 Report
a) Improve chapter 3 "Results and Discussion" describing the following topics: e.g. IF the resulting model learning guaratee the consistent and objective estimates and IF these are accurate enough to support decision making in the agricultural sector (policy makers) and to allow operational use for the stakeholders (farming precision).
b) After the "Conclusion" insert a chapter "General discussion and outlook" covering the following items: i.e. Selecting the most accurate pixel classification model and relevant input features; i.e. Effectively evaluation the classification performance; i.e. Reccomandation for an improved version of the machine learning used; i.e. Formulating a set of reccomandations for future researchers and appliers similar machine learning.
Author Response
Dear Reviewer, I hope you’re doing well.
Thank you very much for the time dedicated and for the relevant comments. Regarding remaining English language issues, we’ll submit the article for English editing after the final review.
As per your comments about the article:
- We clarified the focus and the benefits related to the classification of agriculture row crops in Brazil and the achievements in contrast to other publications that addresses the same objective.
- We made a separation between results, discussion and conclusion. In particular, bringing the achievements to evidence.
- We clarified the Methods, expanded the discussion, and the conclusion toward a more robust explanation
We hope these improvements are able to evidence the contributions of the paper.
- In particular, to your comments
- The results are well supported by a large amount of data, five data sets with 17.000 Pixels Time Series (PTS) were extracted from each studied are. This amount is good enough to give us 99%, confidence level of ±1%. This confidence level is now highlighted in the methods. At this confidence level, we expect that the results can support the sector and that the information is operational for precision farming
We expanded the conclusion with the suggested items: i) most accurate classification model and relevant Input features; ii) effective evaluation of the performance; iii) recommendations for future researches

Reviewer 2 Report
The authors have proposed an approach for agriculture areas classification using remote sensing data by incorporating the similarity metrics. They claimed that this incorporation of similarity metrics is the major novelty.
Frankly speaking, I can hardly understand the manuscript due to its chaotic organization, poor English, and unclear presentation. For instance,
- what is the purpose of Section 2.3 ? I can understand nothing. What is divided here as agriculture and other ? There are many other similar examples as well.
- In line 33, the performance of 96% is given, and it is conflict with the state of the art presented in line 26.
- The relation between values 73,797,057 and 5,184,813 in line 102 and 103 is unclear. And how are they correlated with the one 67,547,537 in line 108 ?
- Table 1.1 is not traceable. What is the relationship between column 2 and 3? What is the relationship between row 1 (Brazil) and the remaining rows? What are these values referring to ? What is "ha" ?
- The Table in line 117 is missing.
The authors have to thoroughly revise their manuscript in both content and language.
Author Response
Dear Reviewer, I hope you’re doing well.
Thank you very much for the time dedicated and for the relevant comments. Regarding remaining English language issues, we’ll submit the article for English editing after the final review.
As per your comments about the article:
- We clarified the focus and the benefits related to the classification of agriculture row crops in Brazil and the achievements in contrast to other publications that addresses the same objective.
- We made a separation between results, discussion and conclusion. In particular, bringing the achievements to evidence.
- We clarified the Methods, expanded the discussion, and the conclusion toward a more robust explanation
We hope these improvements are able to evidence the contributions of the paper.
- In particular, to your comments
- what is the purpose of Section 2.3 ? I can understand nothing. What is divided here as agriculture and other ? There are many other similar examples as well
The objective was to put in evidence the main types of crops covered by the experiment (soybean, corn, cotton, bean, potato, manioc and tomato). In particular, we can have two or even three seasons per year in Brazil; The chart has been improved to give the proper emphasis and for clarification.
- In line 33, the performance of 96% is given, and it is conflict with the state of the art presented in line 26
You’re right, this citation isn’t in the right place, has been moved to the discussion section. The performance accuracy distinguishing corn from soybean is not the goal of this article. In the discussion section we present Cai2018 as one of the processes that rely on two stages classification, in other words, depended of pre classified areas.
- The relation between values 73,797,057 and 5,184,813 in line 102 and 103 is unclear. And how are they correlated with the one 67,547,537 in line 108
You’re right, clarification implemented. The numbers related to agriculture in Brazil are presented in two groups: Planted (73,797,057) area and harvested (67,547,537)area. Planted area is the total area not taking in consideration that some regions have two or three seasons. Harvested area take in consideration the area, doesn’t take in consideration how may seasons per year. The area 5,184,813 refer to permanent crops that are not part of this study.
- Table 1.1 is not traceable. What is the relationship between column 2 and 3? What is the relationship between row 1 (Brazil) and the remaining rows? What are these values referring to ? What is "ha"
You’re right, clarification implemented. The column 2 shows the total area of the country and then the total area for each state. Column 3 presents the harvested area for Brazil and for each state. Therefore, row 1 is the area for Brazil (country) and the following ones are the area per state.
What is "ha": hectare (ha) - 10,000 square meters. The correct type is: eg. 20 ha. In Brazil official agriculture data are presented using this scale.
- The Table in line 117 is missing.
Fixed

Reviewer 3 Report
The work is well presented and interesting, although it is not very new as it deals with a topic on MODIS that has been worked on for years. The wording is generally correct, although many details have been identified as needing improvement, which are noted below.
The wording of sentences should be improved when integrating the work of an author. The sentence should not start with "[20] explored the synergies ..." but "Other works have explored the synergies ...". So the quote goes at the end of the sentence, not the beginning. It is suggested that the wording of the texts be revised in this sense.
When several bibliographic citations are referenced, such as "[24], [25], [12,14], [13,16], [15] and [20]", the correct format is to incorporate all of them in one such as "[12-16, 20, 24, 25]".
The units should be separated from the figures, for example 20ha should be 20 ha.
Figure 1 and figure 7. The position of the north and the scale are missing. In figure 7, the text of the graph's title should be at the bottom of the figure.
Paragraph 1.2. and other similar cites. Web page references should be incorporated like other bibliographic citations, not as footnotes.
Line 117. Review table number.
Paragraph 2.5. abbreviations TIFF, HDF and CSV should be presented correctly in the text, not as footnotes.
Line 236. Check the table number.
Line 251. Check the decimal format.
Line 257. Incomplete sentence "More over..."
Line 390. Check wording.
The discussion of the results obtained should be improved in comparison with other previous work done in Brazil with MODIS such as:
- Wavelet analysis of MODIS time series to detect expansion and intensification of row-crop agriculture in Brazil
- Classifying multiyear agricultural land use data from Mato Grosso using time-series MODIS vegetation index data
- Classification of MODIS EVI time series for crop mapping in the state of Mato Grosso, Brazil
- Mapping soya bean and corn crops in the State of Paraná, Brazil, using EVI images from the MODIS sensor
- Assessment of MODIS LAI retrievals over soybean crop in Southern Brazil
Also can incorporate references and comparisons with other similar works elsewhere.
Author contributions, please change author and number by the surname or initials.
Figure A1 is not cited in the body of the text. The arrows indicating north should all be the same size. The grey scale to the right of the subfigures is not explained in the figure caption.
About the references list:
Reference 6. Check author formatting and remove year in parentheses.
Reference 8. Check capitalization in initials.
Reference 10. Delete pp.
Reference 15. Check the writing of the title of the article.
Reference 17. Delete (Ny).
Reference 26, 40, 41, 46. Review.
Reference 35. Check DOI.
Author Response
Dear Reviewer, I hope you’re doing well.
Thank you very much for the time dedicated and for the relevant comments. Regarding remaining English language issues, we’ll submit the article for English editing after the final review.
As per your comments about the article:
- We clarified the focus and the benefits related to the classification of agriculture row crops in Brazil and the achievements in contrast to other publications that addresses the same objective.
- We made a separation between results, discussion and conclusion. In particular, bringing the achievements to evidence.
- We clarified the Methods, expanded the discussion, and the conclusion toward a more robust explanation
We hope these improvements are able to evidence the contributions of the paper.
- In particular, to your comments
- The quotes are now at the end of the sentences, not in the beginning.
- The correct citation format has been incorporated;
- units have been separated from the figures;
- Maps Fixed
- Web Citation Fixed
- Table Fixed
- Comments were Incorporated to the text
- points 8 to 11 Fixed
- The proposed articles are important ones and we understood your proposal of including local references as relevant. With this in mind we included TWO of the suggested ones to the discussion to contrast results. From the proposed ones:
- Classifying multiyear agricultural land use data from Mato Grosso using time-series MODIS vegetation index data. 2013 Accuracy 80% and Kappa values were above 0.60
- Classification of MODIS EVI time series for crop mapping in the state of Mato Grosso, Brazil – 2010 – Accuracy 95%and Kappa values 0.98
- We have also included the N arrow
- Author name Fixed
- Figure A1 – Fixed
- references list checked

Reviewer 4 Report
The work "Similarity metrics enforcement in seasonal agriculture areas classification" focuses on a topic of large interest, dealing with the improvement of agricultural land classification based on temporal variation of vegetation index.
The work describe a ML based approach aimed at enhancing the classification of seasonal agricultural areas using the EVI MODIS product while optimizing the computational process bt the use of time-series similarty metrics.
The objective of work should be stated more clear, indeed it seems that EMBRAPA provides a general classification of agricultural areas and it does not allow to discriminate between perennial and annual (temporary) crops and to have information about the number of crop per years, and that the authors want implement a new method to improve this classification. Maybe a clear sentence about the objective would help the reader.
Conceptually, I disagree with use of term phenology in this work. Indeed, phenology mainly concerns with the timing of specific stages of growth and development in the annual cycle of crop, or in general of a specie. I understood that the interpretation of EVI dynamics was used to detect the number and the time of occurrence of possible growth period within the pixel time series, without considering a classification per growth stages. Thus, I suggest to use the term growing season instead of phenology.
I also suggest to improve sentence of lines 140-141 “The agriculture class areas are those where we expected to find a high level of EVI values dynamics, which is the purpose of this study, and is also compatible with the objective of the sensor selected”. What is the meaning of "EVI values dynamics"? "how is it computed"?
Moreover is not clear to me if the contrast dataset used for validation is the IBGE census or the EMBRAPA scene maps or both, and in which ways are used. I’m sorry but this passage it is really not so clear, and maybe author should be improve the methodology description.
For validation, why a 5 m resolution was chosen? is it that of the RapidEye Sensor Scene Maps? please specify.
Overall, the methodology is not always clear, despite several information are provided. All captions are really poor and incomplete.
Please check for typos
For examples
line 88: accurate instead of Accurate
Line 118: phenological instead of phonological
Line 119: requires instead of require
Table 3.1. Please improve the caption: Results of accury classification per area (GO = ...., MG=....)
Figure 6: please remove c)
Line 234: according instead of accord
Line 257: please check More over ..., maybe the sentence is incomplete or it is a typos
Author Response
Dear Reviewer, I hope you’re doing well.
Thank you very much for the time dedicated and for the relevant comments. Regarding remaining English language issues, we’ll submit the article for English editing after the final review.
As per your comments about the article:
- We clarified the focus and the benefits related to the classification of agriculture row crops in Brazil and the achievements in contrast to other publications that addresses the same objective.
- We made a separation between results, discussion and conclusion. In particular, bringing the achievements to evidence.
- We clarified the Methods, expanded the discussion, and the conclusion toward a more robust explanation
We hope these improvements are able to evidence the contributions of the paper.
- In particular, to your comments
- Suggestion accepted : Growing season
- Modified
- Modified
- Correct, the 5 m resolution is Rapid eye Sensor Scene Map. The follow explanations have been incorporated to the discussion:
It's important to highlight that in Brazil we have summer crop season which is in general rainy and cloudy. Moreover, more than 90% of the production is rainfed. Accord to [45] (Barreto2014), less than 40% of temporal images are useful for analysis. From this context, in this research, we bring the need for the combination of high temporal resolution with high spatial resolution for validation, MODIS and RapidEye, respectively.
- Improved
- Corrected the all the appointed errors.

Round 2
Reviewer 1 Report
No additional efforts are requested.
The article now as rewieded is robust!
Author Response
Dear Reviewer,
Thank you for your comments. We have improved references format and implemented small changes to clarify the method.
Please find final version attached
Best
Marcio

Reviewer 2 Report
I have gone through the revision. Although the authors have tried to answer my questions and improved somehow their manuscript, there are still many open points to be clarify in their method.
I am a reviewer with engineering background. Thus, the algorithm namely the approach used in this paper has to be very clearly and rigorously explained. In this revision, the authors add more details to explain the Figure 2. It is good. But the more important one in Figure 3 is still very abstract. For instance, after similarity metrics calculation, what is exactly done to verify the data ? What is then the outcome ? How the training data set is incorporated into the processing chain ? As for me, training is a separate phase, which is normally done beforehand. Further, the authors do not even present the similarity metrics.
I am convinced with this revision.
Author Response
Thank you for your comments. We have implemented changes to clarify the method and improved references format as well.
In particular, to your topics: we improved Figure 3 explanation and text ordering as below:
- In regard to similarity metrics, What is then the outcome? Answer: The whole data is pre-classified into two groups Line 153 – 154
- How the training data set is incorporated into the processing chain?Answer: Each experiment used 20% of data set for training – Line 155 – 156
- As for me, training is a separate phase, which is normally done beforehand. Answer: Since the goal was to validate the data with the similarity metrics against data without the similarity metrics, we took a portion to train and compared the classification accuracy for the whole.
- Further, the authors do not even present the similarity metrics. Answer: The similarity metrics are presented on lines line 160 – 161
Please find final version attached
Best

Reviewer 3 Report
The manuscript has been reviewed again and it is noted that many modifications have been made. But there are some aspects that still need to be reviewed and modified.
Footnotes are still used on pages 3, 5 and 7 to indicate links to web pages. I have suggested that the editor should decide whether this is correct; in my opinion it is not correct and should be properly cited. In any case, please check that links 3, 6, 7 and 8 do not work and should be corrected.
Table 3.1. Please, indicate in the table caption that GO, MG, MT, PR and SP are the abbreviatures of states. People abroad do not know this.
Figure 7 is very interesting and can be resized to the wide of the page. Please, indicate in the figure caption that MT, GO, MG, SP and PR are the abbreviature of states.
Author contributions, please change author and number by the surname or initials.
Reference list:
Some references are misspelled, and the name of the journal appears, followed by the title of the publication. They are number 24, 40.
References 26 and 45 should be written correctly, as they are now in capital letters.
Reference 18 is incomplete and the name of the journal is missing.
Reference 22 is incomplete and the journal and pages of the article are missing.
Reference 33 is not known to which document it responds, it seems to be a doctoral thesis; it should be revised.
Reference 34 has an erroneous DOI.
Reference 35 does not correctly indicate the name of the journal.
Reference 36 does not indicate the name of the journal.
Author Response
Thank you for your comments. We have reviewed the references format for all references and also implemented changes to clarify the method.
In particular, to your topics:
1) Reference 6. Check author formatting and remove year in parentheses. Coase1937a Fixed
2) Reference 8. Check capitalization in initials. Coase – Fixed
3) Reference 10. Delete pp. Mulyono2013 pp Fixed
4) Reference 15. Check the writing of the title of the article. Cai2018 Fixed
5) Reference 17. Delete (Ny). Fixed
6) Reference 26, 40, 41, 46. Review. Fixed
7) Reference 35. Check DOI. Fixed
8) Footnotes were removed and web links properly incorporated to the references. Fixed
9) We have checked all the links, including 3, 6, 7 and 8
10) Table 3.1. Please, indicate in the table caption that GO, MG, MT, PR and SP are the abbreviatures of states. Fixed
11) Figure 7 is very interesting and can be resized to the wide of the page. Please, indicate in the figure caption that MT, GO, MG, SP and PR are the abbreviatures of states. Thank you. Improved!
11) Author contributions, please change author and number by the surname or initials. Fixed
12) Some references are misspelled, and the name of the journal appears, followed by the title of the publication. They are number
a) 24 -Furtado, L.F.d.A.; Francisco, C.N.; de Almeida, C.M. Análise de imagem baseada em objeto para classificação das fisionomias da vegetação em imagens de alta resolução espacial. Unesp Geociências; UNESP., Ed.; UNESP: São Paulo, 2013; Vol. 32, pp. 441–451. Fixed
b) 40 - Cai, Y.; Guan, K.; Peng, J.; Wang, S.; Seifert, C.; Wardlow, B.; Li, Z. A high-performance and in-season classification system of field-level crop types using time-series Landsat data and a machine learning approach. Remote Sens. Environ. 2018, 210, 35–47. doi:https://doi.org/10.1016/j.rse.2018.02.045. Fixed
c) 26 and 45 should be written correctly, as they are now in capital letters. Fixed
d) Reference 18 is incomplete and the name of the journal is missing. Fixed
e) Reference 22 is incomplete and the journal and pages of the article are missing. Fixed
f) Reference 33 is not known to which document it responds, it seems to be a doctoral thesis; it should be revised. Fixed
g) Reference 34 has an erroneous DOI. Fixed
h) Reference 35 does not correctly indicate the name of the journal. Fixed
i) Reference 36 does not indicate the name of the journal. Fixed
Please find final version attached
Best
Marcio
